

# Attitudes towards communication and perceived self-efficacy in nursing students: a longitudinal observational study

Rocío Juliá-Sanchís[1,2], Silvia Escribano[1,2], Juana Perpiñá-Galvañ[1,2], Sofía García-Sanjuán[1,2], María Sánchez-Marco[1,2] and María José Cabañero-Martínez[1,2]

[1] Department of Nursing, University of Alicante, Alicante, Spain
[2] Alicante Institute of Health & Biomedical Research (ISABIAL), Alicante, Spain

## ABSTRACT

**Background**. Communication is recognised as a critical component of all nursing interventions. For nurses to be able to communicate effectively, they need to develop communication skills during their training. Despite this recognition, there is still a lack of consensus about where and when in the syllabus this content should be covered, and how much time should be devoted to this competence, resulting in a inadequate and inconsistent training. Consequently, students develop negative or positive attitudes towards communication. The aim of this study was to analyse the evolution of attitudes towards communication and perceived self-efficacy in a cohort of undergraduate nursing students.

**Methods**. We conducted a prospective longitudinal observational study with three measurement points. Attitudes towards communication and self-efficacy were measured until the 2022–2023 academic year.

**Results**. Participants included 131 undergraduate nursing students with a mean age of 20.44 years ($SD = 6.08$). The scores for attitudes towards communication were not linear. Baseline scores were higher than those recorded at the second assessment, and then scores increased again after the training. Scores for perceived self-efficacy increased progressively over the course of the training programme.

**Conclusion**. Attitudes towards communication and perceived self-efficacy do not intrinsically improve with age, course progression or clinical experience. When specific training in communication skills is not provided, students perceive their communication skills to be moderate and regard communication as a clinical competence of limited relevance . However, after receiving specific person-centred training in their final year, students perceived their ability to be highly in what is a very relevant competence in the healthcare context.

# INTRODUCTION

Person-centred care is a healthcare paradigm that prioritises the individuality, needs, preferences and expectations of patients and empowers them to take an active role in

Corresponding author
María Sánchez-Marco,
maria.marco7@ua.es

managing their health (*Ryan, 2022*). Research has shown that person-centred care is associated with shorter hospital stays, lower readmission rates, better quality of care and greater satisfaction with the care provided (*Rosengren, Brannefors & Carlstrom, 2021*).

To move from paradigm to clinical practice, healthcare professionals need to be skilled in effective clinical communication (*Lee & Son, 2022*), which involves not only clear communication of information, but also the ability to listen, understand and respond appropriately to a person's emotional and physical needs. This approach aims to impart knowledge and influence specific perceptions and attitudes (*Stehr et al., 2022*). The importance of communication is reflected in the nursing theories and models underpinning professional practice. For example, Jean Watson's Theory of Human Caring (*Watson, 2018*) highlights the vital role that communication with both patients and their families plays as part of the care process.

By the late 2010s, a clear picture began to emerge in the literature: the communication training provided to nurses was insufficient to meet the demands of modern healthcare. While communication was widely recognized as a cornerstone of nursing practice, studies revealed that many nurses lacked essential skills in this area (*Sánchez-Expósito et al., 2018*; *Shorey et al., 2018*). Training programmes appeared heavily focused on clinical and technical competencies, with limited emphasis on developing interpersonal skills, teamwork, or effective communication strategies (*Pires et al., 2017*). This gap in training became increasingly concerning when its consequences were considered. Poor communication was linked to longer hospital stays, higher healthcare costs, and ultimately poorer health outcomes for patients (*Agarwal, Sands & Schneider, 2010*). However, despite the urgency to address this issue, there was little agreement on how communication training should be integrated into nursing curricula—questions remained about where and when such content should be included and how much time should be dedicated to it (*Ruiz-Rojo et al., 2022*).

These challenges were further exacerbated by the attitudes of nursing students themselves. As posterior evidence showed, limited emphasis on communication training often led to the development of negative attitudes towards its importance, contributing to a cycle of poor communication and suboptimal patient care outcomes (*Sanchis-Giménez et al., 2023*). Conversely, students who held positive attitudes towards communication were more engaged in their learning, invested more time in honing these skills, and applied them effectively in clinical practice (*Givron & Desseilles, 2021*). Given these findings, the need to not only enhance communication training but also address and shape nursing students' attitudes towards communication became increasingly apparent (*Chou, Tai & Chen, 2023*).

It is worth noting that there is a paucity of research in the nursing field that has looked at this issue in any depth, with most existing studies being cross-sectional in nature. Recent studies by *Giménez-Espert et al. (2021)* and *Sanchis-Giménez et al. (2023)* examined attitudes towards communication in nursing education, revealing differences across student levels and relationships with factors like emotional intelligence and social skills. However, these cross-sectional studies fail to capture how attitudes evolve over time, particularly during clinical training. However, in the two longitudinal studies identified,

there was a worrying trend of declining attitudes towards communication as students progressed through their training, particularly in the clinical setting (*Givron & Desseilles, 2021*; *Rosenbaum, 2017*). This observation suggests that there is an overriding need for longitudinal research to shed light on these changes over time.

There is also a need to tackle the lack of confidence students have in their ability to communicate with patients (*Cannity et al., 2021*). This is due to a number of barriers to effective communication, such as difficulty initiating or maintaining conversations (*Lin et al., 2017*). This study seeks to fill a gap in the literature by incorporating the dimension of perceived self-efficacy in therapeutic communication into the undergraduate nursing programme. Essentially, the aim is to explore students' perceptions of their ability to successfully complete a task (*Bandura, 1995*; *Doyle et al., 2011*). Self-efficacy, grounded in social cognitive theory (*Browne, 2023*; *Skoglund et al., 2018*), plays a significant predictive role in students' motivation, learning and performance (*Escribano et al., 2022*).

The purpose of this article is to provide a comprehensive analysis of how attitudes towards communication and perceived self-efficacy evolve in a cohort of nursing students over the course of their training. The lack of longitudinal research in this area adds weight to the importance and originality of this study, which provides valuable insights that may inform future teaching strategies and better equip students for patient-centred professional practice.

## METHODS

### Design and participants

This is a prospective longitudinal observational study in a cohort of Spanish nursing students, with follow-up at three different points in time: T1, in the first year of the undergraduate nursing degree during the academic year 2019-2020; T2, at the beginning of the fourth academic year in 2022-2023; and T3, at the end of the fourth academic year.

Eligible participants were nursing students from the 2019-2020 cohort at the University of Alicante Bioethics Committee, enrolled in the subject "Support Roles", which is part of the first academic nursing syllabus. Of this total sample (182), 131 individuals (71.98%) agreed to participate in the study and were included in the data collection process. A total of 120 subjects responded at T2 (91.60%) and 106 at T3 (88.33% of those at T2). The percentage of baseline respondents who completed all stages of the study was 80.92% (106/131).

### Training in competences related to communication skills

Courses designed to develop specific competences in clinical communication skills are included in the University's nursing syllabus (*University of Alicante, 2019*). The first academic year subject, "Support Roles" (six European ECTS credits = 150 h), covers competences in basic communication skills and strategies for clinical practice, and is approached from a theoretical perspective. During the second and third years, there are no theoretical subjects specifically focusing on these competences, but they are included as a cross-cutting competence during 1,650 h of clinical placements (66 ECTS credits). Specific communicative skills for managing complex clinical situations are developed in the fourth

year of the programme in the following subjects: "Nursing Community Intervention, Mental Health, Psychiatry and Ethics" and "Nursing Care for Chronic Conditions, Dependence, Geriatrics and Palliative Care", worth nine and six ECTS credits, respectively (see our syllabus at: https://web.ua.es/en/grados/grado-en-enfermeria/curriculum.html#Plan-2). These subjects cover content related to communication skills from a theoretical and practical perspective using high-fidelity simulation with a standardised patient (SPs) over eight sessions, each lasting two hours and 30 min. Each session is divided into two parts, each focusing on a different clinical case scenario. The structure of each case includes distinct stages: peer-led theoretical explanation (10 min), scenario simulation (5 min per subgroup), and debriefing (25 min) (*Juliá-Sanchís et al., 2025*). This design promotes a balance between active and reflective learning, enabling students to concentrate on the essential aspects of clinical interaction while preventing cognitive and physical fatigue by avoiding unnecessarily prolonged simulations. From the perspective of the SPs, their involvement in the first clinical scenario consists of a 5-minute simulation followed by a 25-minute rest period (*Juliá-Sanchís et al., 2025*). Then, they repeat the same clinical case and rest again. This cycle is then repeated for the second clinical scenario. Thus, across the two-and-a-half-hour session, SPs actively participate for 20 min, ensuring high performance and safeguarding their well-being (*Hamilton, Molzahn & McLemore, 2024*). Furthers details of the simulation using a standardised patient can be found in *Escribano et al. (2021)* and *Cabañero-Martínez et al. (2021)*.

## Variables and instruments

Sociodemographic variables and variables related to communication skills were collected at the beginning of the study. These included gender (men/women), age (open question) and nationality (Spanish/other).

Attitudes towards communication were evaluated using the Spanish adaptation (*Escribano et al., 2021b*) of the Attitudes Towards Medical Communication Scale (*Langille et al., 2001*). This unidimensional tool comprises 11 items rated on a five-point Likert scale from "strongly disagree" (1) to "strongly agree" (5). The total score ranges from 11 to 55, with higher scores indicating a more favourable attitude towards communication. The Spanish version demonstrates satisfactory internal reliability with a coefficient of 0.75 (*Escribano et al., 2021b*).

Self-efficacy in communication skills was measured using the Spanish edition by *Escribano et al. (2022)* of the Self-Efficacy in Communication Skills Scale (SE-12) by *Axboe et al. (2016)*. This scale is unidimensional and comprises 12 items assessed on an 11-point Likert scale, ranging from "not at all confident" (1) to "completely confident" (10). The total score ranges from 12 to 120, with higher scores reflecting greater confidence in communication abilities. The original version has a strong internal consistency (Cronbach's alpha = 0.95) (*Axboe et al., 2016*), and the Spanish adaptation demonstrates an internal consistency of 0.94 (*Escribano et al., 2022*).

## Procedure

This study was conducted in accordance with the criteria of the Declaration of Helsinki and the European Union Standards of Good Clinical Practice and was approved by the

University of Alicante Bioethics Committee. Students were informed that their participation in the research was voluntary and that they had the right to withdraw at any time. While the academic activity was mandatory, participating in the study would not affect their grades. No incentives were offered to encourage continuous involvement throughout the research phases. All participants signed an informed consent form. The information collected was treated as confidential and was retained solely by the researcher responsible for conducting the analyses.

The initial data collection took place in the first year of the nursing degree, prior to the start of theoretical training in basic communication strategies in the healthcare context. This involved collecting sociodemographic information and administering the scales assessing attitudes towards and self-efficacy in communication. The attitude and self-efficacy variables were then reassessed in the fourth year of the degree, before students began specific training in communication skills in complex contexts in the 2022–2023 academic year and afterwards.

Data were collected at the three time points using an electronic questionnaire created in Google Forms. The questionnaire was distributed *via* University of Alicante Bioethics Committee, the institution's intranet portal, to students enrolled in subjects covering communication skills in the first and fourth years of the nursing degree. At each stage, the form provided detailed information about the study, explicitly asked for informed consent, emphasised the voluntary nature of participation in the research, offered the option to withdraw from participation at any time without justification or consequence and outlined the protocol for handling information. To maximise the response rate, time was made available to complete the questionnaire during in-person training sessions at time points T1 and T2. An announcement was also posted on the intranet portal with a link to the questionnaire and an appeal for participation. At time point T3, an announcement with the link to the online questionnaire was sent out after the fourth-year theoretical subjects had been completed. This timing coincided with the students' practicum, which took place in healthcare centers distributed across the entire province. To maximize participation, three additional reminders were sent at one-week intervals.

## Data analysis

Analyses were performed using SPSS v.26 software (IBM Corp., Armonk, NY, USA). Descriptive analyses were carried out for both sociodemographic and outcome variables. Significant differences in baseline scores were analysed using the non-parametric Mann–Whitney U test for the gender variable and Spearman's correlation test for the age variable, after the variables had been checked for possible violations of the normality assumption.

A repeated measures analysis of variance (GLM ANOVA) was used to compare scores, with the sociodemographic age variable included as a covariate where significant differences were observed at baseline. For all repeated measures analyses, the sphericity assumption from the univariate analysis was used unless Mauchly's W-test indicated that the sphericity assumption had been violated, in which case the Greenhouse-Geisser statistic was used. *Post-hoc* comparisons between the time points were made using the Bonferroni test. The effect size was reported using the $\eta^2$ statistic (*Cohen, 1992*).

We analysed whether the loss of subjects over the course of the study was associated with any of the dependent variables, attitudes towards or self-efficacy in communication, using a difference in means with the Mann–Whitney U statistic. No statistically significant differences were found for any of the variables analysed (attitude: $U = 1{,}266$; $p = 0.72$) (self-efficacy: $U = -1{,}310.5$; $p = 0.93$).

## RESULTS

A total of 131 subjects completed the baseline questionnaire. Of these, 85.5% were women ($n = 112$). The mean age was 20.44 years ($SD = 6.08$, range = 18–51) and 90.8% ($n = 119$) were of Spanish nationality. Similar sociodemographic characteristics were observed at T2 ($n = 120$) and T3 ($n = 106$), with no significant changes in gender distribution, mean age, or nationality.

The baseline assessments revealed high scores for attitude ($M = 50.04$, $SD = 3.87$) and moderate scores for self-efficacy ($M = 82.27$, $SD = 18.27$). No statistically significant gender-based differences were observed for the variables attitude (Mann–Whitney $U = 820$; $p = 0.11$) and self-efficacy (Mann–Whitney $U = 956.5$; $p = 0.48$). However, a statistically significant association was found between age and attitude ($Rho = 0.25$, $p < 0.005$) at baseline, whereas no significant association was found between age and self-efficacy ($Rho$ 0.10, $p = 0.25$).

When the scores were compared (Fig. 1), repeated measures analysis revealed statistically significant differences between the scores obtained for the attitude variable ($F = 33.25$, $df = 1.65$, $p < 0.001$), indicating a large effect size ($\eta^2 = 0.24$). *Post-hoc* analyses using Bonferroni correction showed that the scores for attitudes towards communication were higher in the first year ($M = 50.09$, $SD = 3.57$) than at the beginning of the fourth year of the nursing degree ($M = 47.87$, $SD = 2.32$) (Diff (1-2) = 2.22, $p < 0.001$). Scores were higher at the end of the training ($M = 52.83$, $SD = 5.09$) than at the two previous assessment points ((diff (2-3) = $-4.96$, $p < 0.001$); (diff (1-3) = $-2.75$, $p < 0.001$)).

When the scores were compared (Fig. 2), repeated measures analysis revealed statistically significant differences between the scores obtained for the self-efficacy variable ($F = 35.59$, $df = 1.79$, $p < 0.001$), with a large effect size ($\eta^2 = 0.25$). *Post-hoc* analyses using Bonferroni correction showed that scores for self-efficacy in communication in the first year ($M = 82.43$, $SD = 17.83$) remained stable at the beginning of the fourth academic year ($M = 86.39$, $SD = 14.91$), with no statistically significant differences (Diff (1-2) = $-3.95$, $p = 0.06$). At the end of the fourth academic year ($M = 97.04$, $SD = 14.56$), students perceived their self-efficacy to be higher than at the two previous points in time ((diff (2-3) = $-10.65$, $p < 0.001$); (diff (1-3) = $-14.60$, $p < 0.001$)) (Table 1).

## DISCUSSION

This study looked at the evolution of attitudes towards and self-efficacy in communication skills in a cohort of students enrolled in an undergraduate nursing degree programme.

Our findings indicate that attitudes towards communication evolve in a non-linear fashion over the course of the degree, with a decline in attitudes towards communication
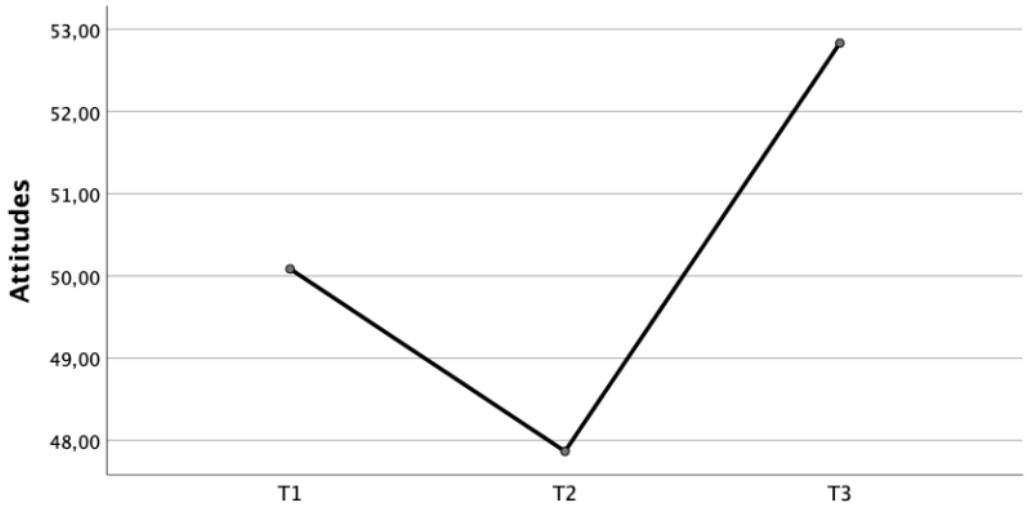

**Figure 1** Longitudinal evolution of attitude scores.

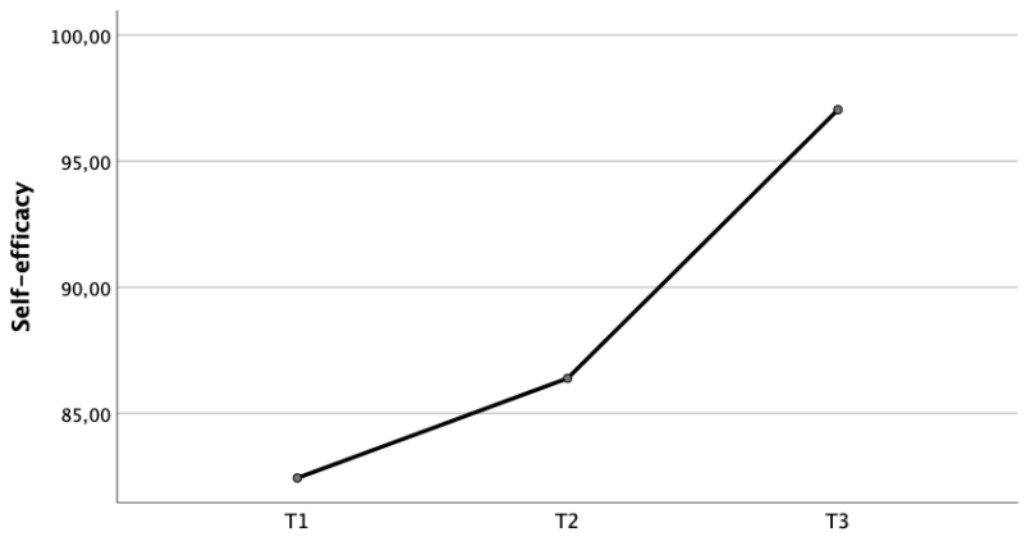

**Figure 2** Longitudinal evolution of self-efficacy scores.

at the beginning of the fourth academic year, followed by an increase at its conclusion. This pattern aligns with prior research involving nursing students, both nationally (*Giménez-Espert et al., 2021*) and internationally (*Kaplonyi et al., 2017*; *Smith et al., 2018*), as well as studies in medical students (*Givron & Desseilles, 2021*; *Ruiz-Moral et al., 2021*). These studies observed declining attitudes towards communication as students progressed through their studies. This could be attributed to experiences of communication that are increasingly negative, challenging and demanding (*Giménez-Espert et al., 2021*; *Ruiz-Moral et al., 2021*), as well as clinical supervisors who lack the knowledge and skills to effectively teach students about communication in a healthcare context (*Rosenbaum, 2017*; *Patidar*

**Table 1** Comparison of the scores of the variables collected in the three items throughout the nursing degree studies ($n = 106$).

|  | T1M (SD) | T2M (SD) | T3M (SD) | F | df | p | Effect size [a] | Diff 1-2 | Diff 2-3 | Diff 1-3 |
|---|---|---|---|---|---|---|---|---|---|---|
| Attitudes towards health communication | 50.09 (3.57) | 47.87 (2.32) | 52.83 (5.09) | 33.25 | 1.65 | <0.001 | 0.24 | 2.22[*] | −4.96[*] | 2.75[*] |
| SE-12 | 82.43 (17.83) | 86.39 (14.91) | 97.04 (14.56) | 35.59 | 1.79 | <0.000 | 0.25 | −3.95 | −10.65[*] | −14.60[*] |

**Notes.**

T1, First year of the nursing degree; T2, At the beginning of the fourth course of the nursing degree; T3, At the end of the fourth year of the nursing degree course; Df, Degree of freedom; SE-12, Spanish version of Self-efficacy in Communication Skills; Diff, Differences between the means, calculated *post hoc* with the Bonferroni test.

[a]Effect size calculated by $\eta^2$.

The age variable has been included as a covariate for repeated measures analysis of variance (ANOVA GLM) of the attitude variable.

[*]$p > .001$.

*et al., 2024*). As we will discuss in more detail below, this negative perception is indicative of the ineffectiveness of the educational process (*Škodová, Bánovèinová & Bánovèinová, 2018*), whether due to features of the syllabus or clinical placements (*Givron & Desseilles, 2021*).

In the analysis of attitudes towards communication, a longitudinal pattern emerged: attitudes showed a significant decline at the beginning of the fourth academic year, followed by an improvement towards the end of the same year. According to our syllabus, this decline was associated with students' challenging clinical experiences and a perceived lack of adequate preparation for these scenarios. During the initial years, students demonstrated more positive attitudes towards communication, which later shifted as they encountered clinical environments prioritizing technical care over patient-centred communication. An analysis of the syllabus revealed three important factors that may adversely affect students' attitudes towards communication. Firstly, it is overwhelmingly biomedical in nature; secondly, there is an emphasis on the acquisition of technical skills, which thwarts the integration of and focus on the benefits of communication skills in everyday practice (*Givron & Desseilles, 2021*); and thirdly, as noted by *Michael, Dror & Karnieli-Miller (2019)*, there is a curriculum gap during the nursing degree where students face complex communication situations during their clinical placements without the proper preparation and educational support (*Giménez-Espert et al., 2021*).

When we drill down into the challenges faced on clinical placements, four aspects emerge. Firstly, from an organisational perspective, the fact that clinical placements are undertaken in predominantly biomedical, technology-based and hospital-centric care settings, where the focus is on the reason for admission rather than the person, leads to the dehumanisation of care (*Valenzuela-Anguita et al., 2019*). Secondly, there is an inconsistency between the approaches taken in academic and clinical settings, which perpetuates the gulf between the two environments (*Givron & Desseilles, 2021*). Thirdly, the wide variation in the behaviour and attitudes of clinical supervisors (who serve as role models for students) is due to their lack of knowledge and skills in effectively teaching communication skills within the healthcare environment. This has a negative impact not only on the way supervisors model behaviour, but also on the quality of the patient-centred therapeutic relationship (*Ammentorp et al., 2014*; *Lin et al., 2017*; *Rosenbaum, 2017*). Finally, academic and clinical supervisors tend not to devote specific time or material to communication, despite it being an assessed component of clinical placements (*Giménez-Espert et al., 2021*).

In terms of self-efficacy in communication, our findings suggest that neither age nor the mere experience of having participated in clinical placements is sufficient to significantly improve self-efficacy. As can be seen, a significant increase in self-efficacy in communication skills is only observed after undergoing specific training in the final year. This observation is supported by *Skoglund et al. (2018)*, who reported similar findings in second-year and final-semester students enrolled in a three-year degree programme, demonstrating that progression in knowledge and skills occurred independently of work experience or age. These findings are important because of the pivotal role self-efficacy plays in the academic and professional performance of healthcare professionals (*Abusubhiah et al., 2023*), mediating between theoretical and practical knowledge, behaviour and clinical experience (*Abusubhiah et al., 2023*). Self-efficacy also influences the acquisition, development and retention of competences (*Bandura, 1995*), making it a decisive factor in coping with challenges. Consequently, students with high levels of self-efficacy set higher goals and display greater tenacity in the pursuit of these goals (*Michael, Dror & Karnieli-Miller, 2019*). However, there is a need for further research into whether higher levels of self-efficacy lead to better overall performance (*Bulfone et al., 2022*).

The implications of our study are evident due to the use of a longitudinal approach. This clearly shows that attitudes towards communication and self-efficacy do not intrinsically improve with age, course progression or clinical experience. However, after receiving targeted person-centred training in their final year, students perceive their ability to be high in what they now consider to be a very important healthcare competence. When communication is treated as a cross-cutting competence in clinical placements, without specific training sessions, students perceive their communication skills to be moderate and view communication as a clinical competence of limited relevance in healthcare. This finding highlights disparities in the perception of communication as a core competence, which can be linked to the curriculum's gaps, students' exposure to complex communication scenarios without proper preparation, and biomedically-oriented syllabuses. However, it is important to note that the phrase "clinical competence of limited relevance" is not meant to suggest that communication is inherently irrelevant. Instead, it reflects the perception of students prior to specific training in communication, influenced by their experiences during placements and the structure of the academic curriculum. This clarion call for reform translates into a pressing need for systemic change at macro, meso and micro levels (*Ryan, 2022*). This entails legislative change at the macro level, institutional and organisational transformation at the meso level, and targeted attention to the individual competences and practices of future healthcare professionals at the micro level (*Ryan, 2022*). Understanding how attitudes towards communication and self-efficacy evolve over time provides a sound basis for curricular design in nursing education. This will help to equip students with effective communication skills that will have a positive impact on their future behaviour (*Ajzen, 2011*).

For this reason, and to promote practical and proactive change, we recommend that a number of elements be considered in the design of the undergraduate nursing syllabus: (1) align curriculum with the person-centred care paradigm (*Ryan, 2022*); (2) systematically
introduce multimodal interventions with increasing levels of fidelity and difficulty (*Molina-Rodríguez et al., 2023*), starting with basic skills and progressing to simulation-based learning (*Lo & Hsieh, 2020*), to improve self-efficacy in communication skills; and (3) provide training and support to clinical supervisors on how to deal with communication skills in the healthcare setting (*Rosenbaum, 2017*).

## LIMITATIONS

One potential limitation of this study is the involvement of students in sending reminders during the data collection process. Although measures were taken to mitigate ethical concerns, such as providing training on ethical research practices and ensuring that reminders were framed neutrally, unintended bias or perceived coercion remains. For instance, the relationship between the sender and recipient could have influenced response rates or the quality of responses (*Resnik, 2018*). Future research could explore alternative methods for sending reminders, such as employing automated systems or third-party personnel, to minimise these risks further and enhance data collection procedures' robustness.

The response rate of 19.08% represents a limitation of this study despite implementing a standardised data collection procedure to encourage retention (*Teague et al., 2018*). Participation rate can be partially attributed to the well-documented 10% dropout rate of students between the 1st and 4th academic years in our faculty, commonly caused by factors such as relocation, changes in field of study, or academic underperformance. Moreover, during the final academic year, mobility scholarships could play a significant role in altering the composition of the student cohort, resulting in an influx of international students and the departure of some of our participants, who were consequently excluded from the study. Also, it is essential to note that T3 coincided with the students' practicum, introducing a challenge to response rates, as students were immersed in their clinical duties and geographically dispersed. Despite our efforts, the response rate was likely affected by the demanding schedules and varying accessibility to online resources during this period.

Other limitation is the simulation was conducted as an integral part of one subject during the final year of the nursing degree, as this is a methodology which is in use and approved in the current syllabus. This precludes the use of an alternative approach, such as an experimental design, with the potential for analysis of its effectiveness in a control group, thereby increasing the internal validity of the study. Experimental designs in the form of clinical trials should therefore be used in the future.

Future research should also evaluate the communication skills acquired and their application in clinical practice, *i.e.,* whether learners can incorporate these new behaviours in the applied context over an extended period of time (*Lo & Hsieh, 2020*). In addition, insight into skills retention will help in the planning of future programmes with a view to reinforcing skills through the nurses' working lives (*Cannity et al., 2021*). These findings are critical for educators involved in the development of communication skills during undergraduate and postgraduate nursing training.

## CONCLUSION

Attitudes towards communication and perceived self-efficacy do not intrinsically improve with age, course progression or clinical experience. In the absence of dedicated training sessions, students perceive their communication skills as moderate and regard communication as a clinical competence of limited relevance. However, after undergoing specific person-centred training in their final year, students perceive themselves as highly capable in a competence that is in fact highly relevant in the healthcare context.

These disparities are attributed to gaps in the curriculum, students being exposed to complex communication scenarios without the proper preparation, and biomedically-oriented syllabuses. Clinical placements pose additional challenges, including a lack of sessions focusing on communication, dehumanising environments with a focus on biomedical issues, and variations in the behaviour and attitudes of clinical supervisors.

We propose such practical measures as aligning the curriculum with the person-centred care paradigm, systematically integrating multimodal interventions with increasing levels of fidelity and difficulty, as well as providing clinical supervisors with training and support to deal with communication skills in the healthcare setting. A holistic and adaptive approach is needed in order to foster positive attitudes and effective communication skills in future healthcare professionals.

### Funding

This work was supported by University of Alicante Institute of Education (No. 5940). The funders had no role in study design, data collection and analysis, decision to publish, or preparation of the manuscript.

### Grant Disclosures

The following grant information was disclosed by the authors:
University of Alicante Institute of Education (No. 5940).

### Competing Interests

The authors declare there are no competing interests.

### Author Contributions

- Rocío Juliá-Sanchís performed the experiments, analyzed the data, authored or reviewed drafts of the article, and approved the final draft.
- Silvia Escribano conceived and designed the experiments, analyzed the data, authored or reviewed drafts of the article, and approved the final draft.
- Juana Perpiñá-Galvañ performed the experiments, prepared figures and/or tables, authored or reviewed drafts of the article, and approved the final draft.
- Sofía García-Sanjuán conceived and designed the experiments, performed the experiments, authored or reviewed drafts of the article, and approved the final draft.

- María Sánchez-Marco conceived and designed the experiments, performed the experiments, analyzed the data, authored or reviewed drafts of the article, and approved the final draft.
- María José Cabañero-Martínez analyzed the data, prepared figures and/or tables, authored or reviewed drafts of the article, and approved the final draft.

## Human Ethics

The following information was supplied relating to ethical approvals (i.e., approving body and any reference numbers):

This study was conducted in accordance with the criteria of the Declaration of Helsinki and the European Union Standards of Good Clinical Practice and was approved by the University of Alicante Bioethics Committee (UA-2018-10-24). All participants signed an informed consent form.

## Data Availability

The research available at the Open Science Framework: Rocío Juliá-Sanchís, Silvia Escribano, Juana Perpiñá-Galvañ, Sofía García-Sanjuán, María Sánchez-Marco, and María José Cabañero-Martínez. 2024. "Attitudes towards Communication and Perceived Self-Efficacy in Nursing Students: A Longitudinal Observational Study." OSF. November 19. osf.io/tyf3j.

## Supplemental Information

Supplemental information for this article can be found online at http://dx.doi.org/10.7717/peerj.19139#supplemental-information.

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
