# Peer review of "Attitudes towards communication and perceived self-efficacy in nursing students: a longitudinal observational study"

_PeerJ, doi:10.7717/peerj.19139_

## Round 0.1 · original submission · Major Revisions

This study, while investigating a relevant topic—the evolution of communication attitudes and self-efficacy among nursing students—presents several issues that require the authors' attention before publication.

Regarding the experimental design, the length of the standardized patient sessions (2 hours and 30 minutes) raises concerns about potential fatigue for both students and patients. It is necessary to clarify whether individual or collective responses were compared across different time points of the study. Furthermore, it's important to specify if incentives were used to ensure continuous participation throughout all phases of the research.

Concerning the validity of the findings, the inclusion of demographic data such as marital status and ownership of computers or cars is not justified, given its lack of clear relevance to the study's objectives. The reference to Rosenbaum (2017) regarding the lack of communication teaching skills among clinical supervisors needs to be supported with more recent evidence. The meaning of "intermediate courses" needs clarification, as does the assertion that higher self-efficacy leads to better overall performance, which requires further evidence. It's crucial to ascertain whether the high dropout rate refers to the study or the course, and to discuss participant attrition, including whether any questionnaires with missing data were excluded.

In terms of the presentation of results, the reviewers suggest displaying the progression of participants at the three data collection points, along with their characteristics. The introduction is too lengthy and needs to be more concise. The concept of "communication" requires further clarification and alignment with the definitions from the National Library of Medicine. The descriptors "communication skills" and "undergraduate nursing" should be replaced with more appropriate terms like "social skills" and "nursing students".

Finally, it is crucial to discuss the ethical implications of designating students to send reminders during data collection. The claim of "clinically irrelevant competence" needs to be clarified as it is not reflected in the findings. The authors should specify the study design with commonly used terms, indicating whether it is prospective or retrospective. It is important to also address potential biases that could have affected the results.

·

Basic reporting

Dear authors,

First of all, I would like to congratulate you on the relevance of the work you have developed. Indeed, the development of communication skills among nursing students is crucial.
Let me say that the article, in general, is well-written and meets the requirements of a scientific paper.
Next, I will proceed with a more detailed review of each component of the article.
Regarding the title of the article, it fulfills its purpose. Indicate the study’s design with a commonly used term in the title or the abstract.
You might consider providing a more detailed description of the methodology. Although it is described as a longitudinal observational study, it is not clear whether it is a prospective or a retrospective study.
Consider revising the following keywords: Communication skills and Undergraduate nursing. These are not recognized health descriptors. You might consider replacing Communication skills with Social Skills and Undergraduate nursing with Students, Nursing.
Regarding the introduction, it effectively explains the scientific background and rationale for the investigation being reported. It presents studies that provide a solid foundation for the need to study the proposed topic.
However, I believe the concept of communication requires further exploration and clarification. It would be important to clarify what the authors mean by communication. If we consider the definitions proposed by the National Library of Medicine, particularly as health descriptors, it seems necessary to provide this clarification.
The objectives are clearly defined.

Experimental design

The study design is clear and well-constructed. It appears to be relevant to the aims of the journal.
Regarding the methods, they are clear and understandable. The eligibility criteria, as well as the sources and methods used for participant selection, are explained rigorously. The methods of follow-up are also described appropriately.
For each variable of interest, give sources of data and details of methods of assessment (measurement).
However, it does not describe any efforts to address potential sources of bias. It would be beneficial to explore this aspect further.
All statistical methods were described.

Validity of the findings

Results:
Regarding the results, they are presented clearly and address the defined objectives. It may be worth considering presenting the participants' progression at the three time points analyzed, as well as their characterization. Additionally, it might be useful to discuss the reasons for participant attrition throughout the study. It would also be important to mention whether all questionnaires were included in the study or if there were any missing data.
In the discussion, the article summarizes the key results with references to the study objectives.
The discussion addresses the limitations of the study.
The conclusions are well-written and appropriately framed within the context of the work.

Additional comments

None.

Reviewer 2 ·

Basic reporting

The introduction currently spans almost four pages; so, it would be best to consider reducing its length to enhance clarity and conciseness.

Experimental design

In the method part, could you please provide information on whether any incentives were implemented to encourage participation at all time points throughout the study?

Validity of the findings

No comment

·

Basic reporting

Line 69 - 116: The study appears to have commenced in 2018, with data collected in 2019 - 2020, 2022-2023, and a second time in 2023. Literature references and insufficient field background/context provided. Further evidence is required, pre-2018, to justify the study.

Data set: The data set appears to contain demographic data irrelevant to the study's aims. "The aim is therefore to analyse the evolution of attitudes towards communication and perceived self-efficacy in a cohort of students enrolled in an undergraduate nursing degree." It is unclear why capturing data on marital status, resides, n_computers, cars, holidays, rooms, dishwashers, and bathrooms was necessary. There is no apparent evaluation or discussion regarding these characteristics in the article. How does having a dishwasher affect the evolution of attitudes towards communication and perceived self-efficacy?

Table 1. In the note, it states T2 fourth course. Should this state fourth year?

Experimental design

Line 146: standardised patients. A session lasting two hours and 30 minutes is very long. Is it usual for a conversation between a student nurse and a patient to be so lengthy? Listening fatigue would concern both the student nurse and the standardised patient.

Lines 176 -181: Was a comparison made between individual or collective responses between T1, T2 and T3?

Validity of the findings

Line 254 – 256: statement indicating that clinical supervisors, lack the knowledge and skills to effectively teach students about communication in a healthcare context (Rosenbaum, 2017). This paper is dated, and it needs support with more recent evidence.

Line 264: ‘curriculum gap in intermediate courses’, it would be beneficial to clarify what is meant by ‘intermediate courses’.

Line 295: (Doyle et al., 2011) does not specify the need for further research into whether higher levels of self-efficacy lead to better overall performance.

Line 328: high dropout rate. Is high dropout rate from the course or the study?

Line 331 – 332: “One possible strategy would be to designate key students to act as liaisons to send reminders during the data collection process.” The ethics of this require discussion.

Line 345: ‘clinically irrelevant competence’. This statement is not reflected in the findings or discussion.

Additional comments

Thank you for submitting your paper. This is an interesting and informative study.

---

## Round 0.2 · accepted · Accept

The authors have done an excellent job in revising their study. Their efforts have significantly strengthened the manuscript, addressing key concerns and providing a more thorough and insightful discussion.

·

Basic reporting

No comment.

Experimental design

The modifications reflect the suggested improvements. The authors are to be congratulated.

Validity of the findings

The modifications reflect the suggested improvements. Well done.

Additional comments

The authors are to be congratulated on the improvements they have made to their work. They reflect honesty and commitment. Congratulations.

·

Basic reporting

No comment

Experimental design

No comment

Validity of the findings

No comment

Additional comments

Thank you for taking the time to consider and respond to my comments.